# Understanding General Practitioner and Patient Perceptions Regarding Integration of Non-Pharmacological Interventions in Chronic Non-Cancer Pain Management—A Cross-Sectional Mixed-Methods Study in the RELIEF Project

**DOI:** 10.3390/diseases13020034

**Published:** 2025-01-28

**Authors:** Regina Poß-Doering, Sarina Carter, Sabrina Brinkmöller, Melanie Möhler, Dominik Dupont, Cinara Paul, Marco R. Zugaj, Viktoria Wurmbach, Alexandra Balzer, Michel Wensing, Cornelia Straßner

**Affiliations:** 1Department of Primary Care and Health Services Research, Heidelberg University Hospital, Heidelberg University, 69120 Heidelberg, Germany; sarina.carter@med.uni-heidelberg.de (S.C.); sabrina.brinkmoeller@med.uni-heidelberg.de (S.B.); melanie.moehler@med.uni-heidelberg.de (M.M.); dominik.dupont@med.uni-tuebingen.de (D.D.); michel.wensing@med.uni-heidelberg.de (M.W.); cornelia.strassner@med.uni-heidelberg.de (C.S.); 2Clinic for Palliative Medicine, University Hospital Heidelberg, 69120 Heidelberg, Germany; 3Institute of Health Sciences, University Hospital Tübingen, 72076 Tübingen, Germany; 4Department of General Internal Medicine and Psychosomatics, Heidelberg University Hospital, Heidelberg University, 69120 Heidelberg, Germany; cinara.paul@med.uni-heidelberg.de; 5Department of Anesthesiology, Pain Medicine Section, Heidelberg University Hospital, Heidelberg University, 69120 Heidelberg, Germany; marco.zugaj@med.uni-heidelberg.de; 6Internal Medicine IX–Department of Clinical Pharmacology and Pharmacoepidemiology, Cooperation Unit Clinical Pharmacy, Heidelberg University Hospital, Heidelberg University, 69120 Heidelberg, Germany; viktoria.wurmbach@med.uni-heidelberg.de; 7Institute of Medical Biometry (IMBI), Heidelberg University Hospital, Heidelberg University, 69120 Heidelberg, Germany; balzer@imbi.uni-heidelberg.de

**Keywords:** chronic pain, primary care, general practice, non-pharmacological interventions, bio-psycho-social model

## Abstract

**Background:** Chronic non-cancer-related pain is an independent condition with a multicausal genesis. Guidelines highlight the need for holistic treatment based on the bio-psycho-social model. While prescribing medication is common, it remains unclear how and to what extent non-pharmacological interventions are considered and recommended in general practice pain management. The project RELIEF explored the integration of non-pharmacological interventions in general practices in Germany from both physician and patient perspectives. **Methods:** A mixed-methods study collected data with patients and general practitioners via semi-structured telephone interviews and self-developed questionnaires. Qualitative data were analyzed in a reflexive thematic analysis. Survey data were analyzed descriptively. **Results:** N = 383 questionnaires (n = 131 general practitioners, n = 252 patients) and n = 61 interviews (n = 21 general practitioners, n = 40 patients) were analyzed. Patient and physician perceptions regarding the integration of non-pharmacological interventions differed. Patients felt pharmacological therapy was recommended primarily, applied non-pharmacological interventions based on their own initiative, and were aware of bio-psycho-social interrelations. Physicians perceived that they often recommended physiotherapy and psychotherapy alongside analgesics, and asked about non-pharmacological interventions (79.4%), explained the bio-psycho-social chronic pain genesis (55.7%), and provided information on physical (48.9%) and social (35.9%) activity, relaxation techniques (42%), topical applications (31.9%), and support groups (25.2%). **Conclusions:** The integration of holistic pain management and communication between patients and general practitioners appear to need strengthening.

## 1. Introduction

Chronic pain is a widespread health condition worldwide [1]. In Germany, 23 million people were affected in 2022 [2]. Demographic change with rising life expectancy and an associated increase in multimorbidity favor the prevalence of chronic pain. As a result, providing care for these patients will continue to gain in relevance [3,4].

Pain is considered chronic if it persists or recurs over a period of three months [5,6]. While acute pain immediately follows an event and acts as an endogenous warning system, chronic pain has no apparent physiological benefit [7]. It is considered as an independent condition with a bio-psycho-social multicausal genesis [2,8,9], which poses challenges for treatment [2,9]. Affected patients frequently experience limitations regarding everyday life and physical function [10], in maintaining social contacts and quality of life [11], sleep and anxiety disorders, and depression [2,12]. In addition to personal burden, chronic pain also results in high economic costs [1,12], composed of healthcare system utilization and inability to work [2,11,13].

A valuable framework for the care and treatment of individuals with chronic pain is the bio-psycho-social approach as it takes the whole person into account and considers the body and mind as interconnected entities [14,15]. German guidelines for chronic non-cancer pain (CNCP) management state the need for treatment based on the bio-psycho-social approach, which includes both pharmacological and non-pharmacological interventions [5,16]. This approach is pursued in multimodal pain therapy, though not offered extensively in Germany [17]. Particularly in the outpatient sector, chronic pain management is still considered inadequate [17,18]. General practitioners (GPs) play a key role regarding access to information, care services, and pain treatment. They often have a long-standing relationship with their patients and know their medical and personal history as well as their social situation [19]. Studies show that for the majority of CNCP patients, their GP is the main point of contact for health issues [4,8,17], education, and prevention, and the main prescriber of their medication. These are ideal conditions for the treatment of patients with chronic pain, but the implementation of holistic pain management based on the bio-psycho-social model also poses challenges for GPs [19].

According to guideline recommendations in Germany, non-pharmacological interventions (NPIs) such as structured assessment, pain education, exercise, physiotherapy, relaxation procedures, or psychotherapeutic interventions should be given high priority in chronic pain management [5,16]. However, this is not reflected in studies on pain treatment in primary care. While prescribing of medication is a frequently described intervention, NPIs are hardly reported [4,11,20], and their consideration and integration in outpatient pain management remains unclear [17].

The research project RELIEF (Resource-oriented case management for patients with chronic pain and frequent use of analgesics in general practice) aims to develop and evaluate a case management program intervention for CNCP patients in general practice. The program will regard best-practice recommendations on “assessment and monitoring of CNCP”, “patient education and support”, “promotion of self-care and non-pharmacological interventions”, and “rational pharmacotherapy”. Prior to the development of the intervention, previous patient experiences and GP perceptions of their own treatment approaches were to be explored to gain insights into the current integration of NPI in ambulatory CNCP management. Therefore, the aim of the present study was to explore integration of NPIs in general practice from physician and patient perspectives and to inform intervention development. A pilot study will assess the feasibility of the case management program, and following potentially necessary adaptations, it will be tested in a confirmatory study in 2025.

## 2. Materials and Methods

### 2.1. Study Design and Context

In a convergent parallel mixed-methods design, an explorative cross-sectional retrospective study comprising telephone interviews with GPs and patients with CNCP as well as self-developed surveys was conducted to gain insights into physicians’ and patients’ perspectives on various aspects of pain management (see Appendix A for topics covered in the survey and the interviews). This approach was chosen as it facilitates the inclusion of different perspectives and to obtain a complete picture of the explored field [21]. The questionnaires were designed to identify potential evidence–practice gaps regarding various non-pharmacological aspects of pain management such as structured assessment, education, physical activity, and psychological interventions. Questionnaires were piloted with two GPs and four members of pain-related self-help groups. Completion was calculated to take approximately 20 min. Semi-structured interview guides were developed in a structured approach [22] and focused on gaps identified in the survey to further gain insights regarding aspects relevant for the development of the case management intervention. Using the think-aloud method [23], interview guides were piloted with one experienced GP and one patient. Subsequently, slight changes to wording were incorporated to avoid the overuse of medical terminology (patient questionnaire) and increase the clarity of questions (GP and patient questionnaire). The anticipated interview duration was 30 to 45 min.

This study was approved by the Ethics Committee of the Medical Faculty of Heidelberg (reference S-087/2023). RELIEF is funded by the German Federal Ministry of Education and Research (funding code: 01GY2106).

### 2.2. Recruitment

Survey participants:

Various channels were used for recruitment. GPs received an invitation and a link to participate in the anonymous online survey via the e-mail distribution list of the German Society of General Practice and Family Medicine (DEGAM) in the German federal state of Baden-Württemberg (exact number of recipients not provided). An invitation and link to the survey were also sent to a sample of 500 GP practices located in the Rhine-Neckar region by postal mail.

An invitation and link to the online patient questionnaire were forwarded to chronic pain self-help groups by e-mail via the state-wide self-help umbrella organization and were also distributed via the Instagram account of Heidelberg University Hospital. In addition, paper-based questionnaires were sent by postal mail to a total of n = 3200 individuals. Addresses were obtained through a random sample from resident registers in six selected municipalities in the Rhine-Neckar region (total population around 550,000). The purposive sampling method [24] was used to achieve an approximately equal gender distribution, inclusion of urban and rural areas with less than 10 km or more than 25 km distance to a hospital, and proportionate inclusion of residents without German citizenship. The addressed individuals were asked to complete the questionnaire only if they were affected by chronic non-cancer pain. Given the assumption that older people are increasingly affected by chronic pain [25] but might not be reached sufficiently through digital channels, initially, n = 1200 people over the age of 70 were sent the paper-based patient questionnaire by postal mail. In a second recruitment wave, n = 2000 residents from the 18 to 69 age group received the questionnaire to supplement already gained impressions with the perspective of the younger target group.

### 2.3. Interview Participants

All questionnaires included a call for expression of interest in participating in an interview. Additionally, publicly available addresses of GPs in 71 communities in Baden-Württemberg were randomly selected in a web-based search and n = 900 practices were contacted via e-mail. After expressing interest in participating in an interview, potential participants were sent an information leaflet and a declaration of consent. All GPs and patients who returned the signed consent form were contacted by telephone by the interviewers to provide information regarding the study aim and procedures and to agree on a date and time for the interview. Inclusion criteria for patient participation in the interview included being of legal age, good mastery of German, and pain for more than three months. Only one GP per contacted practice could participate. A compensation of EUR 50 was offered for participation in an interview. All participants gave written informed consent for participation in an interview and analysis of the collected data prior to the interview.

### 2.4. Data Collection

All survey data were collected anonymously between April and August 2023. For the online survey, the tool LimeSurvey was used (hosted on secure servers of Heidelberg University). Paper-based patient survey data were manually added into the tool by support staff. Semi-structured guide-based telephone interviews were conducted and digitally recorded by two junior health services researchers between June and November 2023. There was no personal relationship with the interviewees. Additional notes were taken during and after interviews to record interviewer impressions. The first patient interview was used to pilot the interview guide and no audio was recorded. All qualitative data were collected at the Department of Primary Care and Health Services Research, University Hospital Heidelberg, and stored on secure servers. Prior to patient interviews, socio-demographic data and pain-related information such as pain duration and intensity were collected using a short questionnaire sent by mail. Socio-demographic GP characteristics were collected verbally directly before the interview, or by mailing the questionnaire.

### 2.5. Data Analysis

All audio recordings were pseudonymized and transcribed verbatim using the artificial intelligence software noScribe Version 0.3 [26]. All transcripts were meticulously checked for plausibility and amended where necessary by study team members and student support staff. Transcripts were not checked or corrected by participants. The software MAXQDA 2022 Plus (Release 22.7.0; VERBI Software GmbH, Berlin, Germany) was used for data organization and management. Data analysis followed the approach of Reflexive Thematic Analysis [27,28]. A thematic analysis can be used to take a detailed look at different nuances of a topic (integration of NPIs in pain treatment in this study). Coding of transcripts was conducted independently by three researchers (SC and RPD for patient interviews; SB and RPD for GP interviews) in an inductive–deductive approach and discussed and consented in the study team. After a thorough review of the data and notes, inductive coding led to the identification of overarching themes and sub-themes, which were reflected and revised several times. In addition, and based on topics covered by the interview guides, a deductive step led to the final coding of the entire data. Discussion and consent of all analytical steps in the study team supported the continuous process and self-reflection.

For survey data analysis, all returned questionnaires were transferred into IBM SPSS Statistics 28.0.1.0 (IBM, Armonk, NY, USA), checked for plausibility, and analyzed descriptively. For data visualization, SPSS and Microsoft Excel Version 1808 (Microsoft, Santa Rosa, CA, USA) were used. Only answered survey items were included for analysis; missing values were marked by a specific code. Descriptive statistics were used to (a) characterize the study sample, and (b) assess the perception of current CNCP care by tabulating measures of the empirical distribution. According to the level of variables, means, standard deviations (SDs), and absolute or relative frequencies are reported.

## 3. Results

For the purpose of triangulation, the findings of both parts of this study are detailed sequentially along with themes identified deductively from the qualitative data. Included quotes were translated with due diligence and are presented with an indication of alias (GP = general practitioner, P = patient, Number = number of interview) and transcript position (# followed by number). All findings are presented with reference to two identified key themes: (1) pain treatment approach, and (2) non-pharmacological interventions. Figure 1 shows the main themes and identified relevant sub-themes.

### 3.1. Sample Characteristics

The survey questionnaires were returned by n = 131 GPs and n = 252 patients. Interest in participation in an interview was expressed by n = 50 patients (response rate 19.8% related to n = 252 returned questionnaires) and n = 27 GPs (response rate 3% related to 900 contacted practices). After screening for exclusion criteria and scheduling efforts, a total of n = 61 interviews were conducted (GP n = 21; patient n = 40). All interviews were conducted between June and November 2023. Except for the first patient interview (pilot of interview guide) and two patient interviews where exclusion criteria became apparent during the interview (no CNCP, insufficient knowledge of German), all interviews were included for analysis (n = 58). Mean interview duration was 50:29 min (range 31–88 min). Older patients showed strong interest in interview participation. Most were retired; only 21.4% (survey) and 21% (interview sample) were still employed. Gender distribution was balanced. Table 1 details the socio-demographic characteristics of both samples. 

Patient survey participants reported various pain localizations (neck, head, back, stomach, joints, muscle and tendon, skin). Most of them stated the ability to manage their household (n = 230; 91%), meet family and friends (n = 233; 92%), and cope with their pain in most situations (n = 230; 91%). A total of 67.9% (n = 171) stated they had used analgesics during the last four weeks. In the interviews, patients named several reasons for their pain, including migraines, neuralgia after herpes zoster infections, autoimmune diseases such as rheumatism, multiple sclerosis, and degenerative conditions of the musculoskeletal system. The underlying disease and resulting pain were reported to be accompanied by restrictions in everyday life, such as difficulties in walking and prolonged standing or sitting, regarding physical activities such as cycling and managing their own household. Participants from the 18–69 age group reported pain-related limited performance or even inability to work. Negative effects on social life, sleep, mental health, and quality of life were described.

### 3.2. Pain Treatment Approach

#### 3.2.1. Survey Data

GPs were asked to indicate perceptions about the prevalence of CNCP among their patients, and their treatment approaches and routines. Table 2 details findings regarding frequency-related statements.

In total, 63.4% (n = 83) of the participating GPs indicated they often asked patients with CNCP about social isolation and loneliness, and only 25.2% (n = 31) stated that they rarely or never explained the bio-psycho-social genesis of pain to their patients. They also stated that they assessed unfavorable behavior patterns such as catastrophizing (67.2%; n = 88), and recommended psychotherapy or cognitive behavioral therapy (41.2%; n = 54).

Patients were asked to indicate which disciplines were involved in their CNCP treatment. Table 3 details the respective findings.

More than a third of the patients specified in the survey that they had never been asked to indicate pain intensity using a scale (36.5%; n = 92), describe their pain (35.3%; n = 89), or give their assumptions about pain genesis (35.3%; n = 89). A total of 52.8% (n = 133) indicated they had never received an explanation regarding pain genesis, and 51.6% of the patients (n = 130) indicated that during the course of treatment it had been explained to them that biological, psychological, and social factors contributed to pain. Table 4 describes the patient perception of discussed aspects.

#### 3.2.2. Qualitative Data

All GPs perceived their patients with CNCP as mainly older persons with pain of various etiologies. They described that patients approached them with high expectations regarding pain therapy and they often had to clarify what actually could be expected and agreed on as a realistic therapy goal. Sometimes patients consulted their GP for seemingly unrelated ailments and a CNCP diagnosis occurred incidentally. GPs expressed that they took patients and their CNCP seriously and wrote referrals to specialist treatment, hospital care, and rehabilitation facilities when they saw no improvement or felt their therapy options were exhausted. Opting for specialist treatment was seen as potentially unsatisfactory for patients and GPs because of scarce availability and being a closed system where GPs had no part.

GPs talked about challenges in comprehending patients’ pain intensity, knowledge transfer referring to why medication was prescribed, and patient adherence to actually taking it as prescribed. One GP indicated that challenges might arise because with chronic pain “[…] you have to see what helps at all” (GP18, #12) and pain reduction was “[…] really difficult for patients” (GP08, #4). They also reflected on the volume of prescribed medication and voiced the concern that perhaps less pain medication would be necessary if they had “[…] time to really engage with the patients” (GP10, #4). Regarding NPIs, GPs mentioned they frequently recommend physiotherapy, psychotherapy, and acupuncture alongside analgesics or opioids. Several GPs described that they rarely use structured assessment instruments since they preferred taking time for individual pain assessment in one-on-one conversations. Some were aware of specialists using questionnaires for pain assessment, yet considered it detrimental to the patient–physician relationship.
“[…] when they see a rheumatologist, they are handed a questionnaire, see the physician who looks into it, and perhaps asks two or three questions, examines the patient and then, that is it. And patients feel insufficiently heard, to be quite frank, after answering all the questions. In general practice, our relationship strives on regular dialog.” (GP14, #24)

Some patients reported contentment with their CNCP treatment and felt their GP taking time for them and a continuity in care provision were strong supporting factors for a sense of being “well looked after” (P18, #29). It was described that good CNCP care was characterized by the practitioners taking a patient’s pain seriously and making efforts to achieve pain relief. Information about the cause of pain and what patients could do themselves to alleviate it were also seen as helpful.
“I think, the most important thing is that people who are in pain are informed about where it comes from and what they can do themselves to reduce it. Not just concentrating on the medication, but saying, I’m doing more for my back now, I’m leaving certain things out because they only harm me.”(P03, #246)

Several patients experienced discontent when they felt their pain was insufficiently addressed, interventions did not result in the desired relief, or when they felt they were not taken seriously and left alone. Long waiting periods for specialist or physiotherapy appointments, and the scarcity of such options in rural regions were rated negatively. Few patients perceived that their physicians did not suggest any specific treatment option. Most stated the main therapy option offered was medication. Only a few patients felt they were involved in the decision-making process regarding treatment options. Communication with care providers was seen as important and more helpful than medication. One patient mentioned situations in which one “just needed to get something of the chest” (P01, #179). More discussion of non-pharmacological options, treatment adaptation along the course of pain, and a more holistic treatment approach were considered desirable.
“Yes, it’s good to look at what causes the pain, you really need to look. Is it an organic thing, physical or blood test, that was part of it, but also what else is going on in my life, the personal holistic view. To simply ask and say, yes, something has to be changed.” (P17, #79)

Regarding aspects of bio-psycho-social interrelations, one interviewed GP explained that simply addressing bio-psycho-social factors with patients would already suffice since asking about pain occurrence, work environment, and stress factors would often bring about reflection. Some GPs stated that, usually, CNCP was “[…] partially psychosomatic and most patients are not open to accepting this at all” (GP08, #4). Some patients were perceived to be skeptical and unwilling to consider reflection or psychotherapy. GPs also explained it was difficult to get interested patients started with psychological treatment due to high demand and long waiting times. They were aware of the necessity to choose a sensitive approach to discussing bio-psycho-social interrelations to avoid patients feeling stigmatized, misunderstood, or even pushed away.
“I find it much easier to motivate patients for psychotherapy if I can elaborate with them that the psyche plays a role in pain development and evaluation. […] particularly with older people, it is very difficult to explain why talking about it should help them now. In the end, I would even worry they feel pushed away […].” (GP11, #62)

Almost all interviewed patients were aware of bio-psycho-social interrelations. Some perceived that physicians did not discuss them. One patient described addressing the topic during a GP visit and felt it was “not taken so seriously” (P08, #61). Others stated they were referred to outpatient psychotherapy or the GP generally supported their desire to use this option, but psychotherapists were the ones who provided information on the multicausal pain genesis, not their GP. It was mentioned that during rehabilitation or physiotherapy appointments not only physical but also psycho-social aspects of pain were discussed. Few patients described explicit discussions with their physician and some perceived that the psycho-social situation was rather recorded during medical pain history interviews and not discussed separately.
“[...] of course, when they ask what is the situation otherwise, you already know, now they are asking about the social environment, or now they are just asking.” (P14, #78)

Patients talked about psychological stress due to recurring and persistent pain and its effects. They described worsening of pain with psychological stress, or that pain was easier to bear if they were in a “good psychological position” (P10, #58). One patient linked psychological coping with motivation to turn to NPIs and “somehow try to improve physically” (P19, #59).

The role of the social environment in coping with pain was also discussed. Patients reported negative effects on social relationships and perceived helplessness among family members. Some mentioned they did not want to burden their families and thus put themselves and their pain last. While few participants reported being confronted with a lack of understanding among their friends and acquaintances, for example, regarding restrictions on their activities, their immediate environment was generally perceived as a great source of support.

“[...] if you have someone who stands behind you and absorbs a lot, then that’s actually half the battle to making it more bearable. But people who are all alone, I can hardly imagine how they should or can or have to cope with it all.” (P10, #8)

### 3.3. Non-Pharmacological Interventions

#### 3.3.1. Survey Data

As shown in Table 2, most of the participating GPs stated they often asked patients about their own efforts regarding NPIs, and more than a third of them claimed to often provide information material on available options. Figure 2 details the non-pharmacological interventions GPs stated to verbally inform their patients about in general.

In total, 62.3% of the patients (n = 157) indicated they had been informed about what they could do themselves to relieve pain, apart from taking medication. It was stated that discussed NPIs included physical activity (59.9%; n = 151), psychotherapy (50.8%; n = 128), topical applications (25.0%; n = 63), dietary aspects (28.6%; n = 72), relaxation techniques (22.2%; n = 56), weight reduction (18.3%; n = 46), and changes in thinking (14.7%; n = 37).

Patients were also asked to rate the importance of aspects regarding non-pharmacological therapy options in a potential case management program. Figure 3 details components and ratings.

#### 3.3.2. Qualitative Data

GPs described that they frequently recommended physiotherapy to their patients with CNCP and perceived that about two-thirds of these patients benefitted from it. Some GPs perceived it was difficult for patients to integrate physical activity in daily routines and, therefore, rather recommended professionally instructed physiotherapy or rehabilitation exercises. Most GPs mentioned they recommended relaxation techniques. One GP felt that some patients tried an intervention because it seemingly worked for a neighbor and then some found it did not help them. All participating GPs but one indicated they predominantly used verbal information transfer.

Patient feedback regarding the feasibility of tested interventions generally was regarded valuable for the practice team “because we can pass it on” (GP20, #134). Another GP considered prescribing options for active and passive interventions to be limited and if patients wanted to engage in yoga, they had “to take care of it themselves” (GP02, #88). GPs reported recommending digital apps to engage in relaxation techniques and autogenic training and to also look into programs for CNCP offered by health insurance providers. It was perceived that in many cases, it was difficult for patients to keep engaging regularly when exercising on their own and that group activities might support patients in their efforts because “It’s just no fun alone” (GP13, #48). One GP felt that progressive muscle relaxation was easy to explain and integrate.
“You do not need a lot of room or accessories. […] it has to be feasible in terms of time and not so complicated, because you don’t have time to explain everything to everyone. So, it has to be something you can take home with you.” (GP02, #38)

Patients perceived physiotherapy, acupuncture, psychotherapy, or a combination of these as helpful and supportive to care continuity. Almost all patients stated physiotherapy was a constant and key feature in their pain treatment and they had to “work actively” (P13, #179) there. They described engagement in physical activity such as swimming, cycling, yoga, gymnastics, stretching, rehabilitation exercise, walks, or gardening. Some attributed high relevance to physical activity and felt it was “the very best you can do“ (P11, #205). Dietary changes such as abstaining from meat, gluten, or sugar were also perceived as pain-relieving. Regular use of passive interventions such as acupuncture, heat application, and homeopathic or herbal remedies was found helpful to cope with inflammation. Some pointed out that passive interventions were self-paying options and not covered by health insurance. Few patients mentioned using topical ointments and home remedies. Both active and passive interventions were perceived to provide a reduction in functional limitations.
“[…] I am glad I can go to physiotherapy, and they do the right exercises with me. And I know it helps. Just two months ago I had to take three breaks on my usual walk, and yesterday I did not have to stop at all. Not fast, but it worked, without pain.” (P12, #159)

Patients talked about using psychological techniques for stress reduction and explicitly mentioned muscle relaxation or mindfulness practice. Some detailed dealing with their chronic pain in psychotherapy, in exchanges with similarly affected patients, or via distracting themselves because that felt “simply better than burying yourself in your ailments” (P11, #187). They also expressed that information from the GP practice on NPIs such as strengthening and relaxation exercises, and home and natural remedies would be highly appreciated. Some reported they explicitly demanded prescriptions for acupuncture, physiotherapy, or psychotherapy; others emphasized that they had received encouragement from GPs and physiotherapists to carry on with activities they already pursued. Engagement in active physical exercise was reported to be mainly based on own initiative, or on receiving recommendations from the social environment, psychotherapists, physiotherapists, during rehabilitation programs, outpatient, or in-patient pain therapy, or their primary care physicians. The integration of physical activity into daily routines was felt to be achieved, for instance, through resistance exercise in fitness centers and at home or in online yoga classes. Most patients signaled a high willingness to try different NPIs and did their own research on potential options.
“[…] I did research on options myself to find out what I can do for my arms, the back, about yoga. Are there things that have a better long-term effect? […] there was no one, not my GP [either] who said do this or that.” (P17, Pos. 9)

## 4. Discussion

The aim of this study was to explore the integration of NPIs into CNCP therapy from the perspective of patients and GPs. Participant perceptions indicate that, often, pharmacological pain therapy was offered as a priority. Patients engaged in various NPIs to alleviate pain based on their own initiative, or following recommendations they received from therapists and their social environment. Almost all interviewed patients were aware of bio-psycho-social influences on pain, saw the GP practice as a suitable place for discussing NPIs, and expressed their desire for individually fitting recommendations. GPs felt CNCP was a frequent consultation cause in their practice. They perceived they often asked patients what they did to alleviate pain besides taking medication, informed on NPIs, and entered into agreements with patients regarding their integration in pain therapy.

Restrictions and the negative effects of CNCP in everyday and work activities as well as quality of life as described by participating patients are consistent with prior findings [1,2]. Regarding pain treatment, patients in this study reported heterogeneous experiences. Some felt well looked after; others reported discontent and feeling overlooked. In 2014, a study found that about 67% of participants with disabling CNCP were satisfied or very satisfied with their current pain treatment [29]. Similar findings had been shown in a larger-scale study in 2011 in which, however, about 90% of the patients found that their pain intensity was higher than it should be with successful pain management [30]. The negative perception of not being taken seriously was also considered problematic in an earlier study where about a third of the participants felt their pain was not believed and a quarter stated they were not taken seriously by physicians [11]. Patients’ satisfaction in this present study was not necessarily associated with pain control since they perceived good care was related to other decisive factors, including being taken seriously, a functioning provider–patient communication, and continuity of care.

Chronic pain as a complex condition with significant implications for a patient’s quality of life requires a comprehensive and interprofessional approach to pain management. Thus, healthcare professionals need specific skills to ensure patient-centered care and favorable outcomes, patient safety, effective communication, and efficient care coordination. These skills, including advanced assessment techniques and knowledge of evidence-based options, should continually be updated through training and education to facilitate delivery of the best possible care [31]. The way healthcare professionals deal with patients’ pain can influence further pain development [32]. A restriction to physical factors and thus a lack of consideration of a multifactorial pain genesis can negatively influence its course [17]. An important outcome of non-pharmacological self-management interventions, for instance, is gaining acceptance of having a chronic condition, where the feeling of being believed is essential [33,34]. Psychosomatic and social factors should, thus, be given thorough consideration during the assessment and treatment of chronic pain [25]. Heymanns et al. [8] found in their study that the bio-psycho-social understanding of chronic pain as a distinct condition with a multicausal genesis is still insufficiently disseminated, although 30% of patients showed an awareness of the influence of psycho-social factors. Patients participating in the qualitative part of this present study were aware of interrelations between physical and psycho-social factors and emphasized the importance of a holistic view of CNCP. However, discussion of such interrelations appeared to lack physician consideration since merely half of the patient survey participants indicated that they had received corresponding explanations during the course of their treatment. In contrast, 75% of the GPs stated that they explain the bio-psycho-social genesis of pain. The evidence base regarding the combination of interventions chosen in RELIEF for patients with CNCP in German primary care is still limited. Nevertheless, it is expected that eligible patients will benefit from the RELIEF interventions and that an intensified use of non-pharmacological treatment strategies as well as improved medication management will result in a reduction in pain-related disabilities and further patient-reported outcomes. Findings in this present study delivered valuable input to the intervention development in RELIEF as discussed below and contributed to the shape and contents of educational modules for both GP practice teams and patients that will be tested for feasibility in a small pilot study in 2025.

A previous study showed that patients from different educational levels or socio-economic backgrounds were able to understand educational content on the subject of pain and that this helped them to better understand their chronic pain. However, healthcare professionals underestimated patients’ ability to understand this information [35]. Inadequate information on bio-psycho-social influences on CNCP might result from such differing perceptions and a potentially deficient verbal information transfer. This can be countered by developing target group-specific information material for patients and content reflection in follow-up consultations [19]. Findings in this present study demonstrate the need for a more sustainable provision of information, ideally in different and target group-specific formats enabling reliable opportunities for reflections. Both components were therefore thoroughly considered during development of the RELIEF intervention and will be included in the case management program to facilitate open discussion and foster realistic expectations.

Not only information on chronic pain but also its treatment can be insufficiently oriented towards the bio-psycho-social pain model [8]. Corresponding with recommendations in current German guidelines for the management of chronic pain [16,36,37], patients in this study found an explanation regarding pain genesis and a combination of pharmacological and non-pharmacological interventions supportive, yet many stated they had mainly been offered pharmacological therapy. GPs on the other hand felt they provided verbal information on NPIs such as psychotherapy and physical activity. Physiotherapy was mentioned as a key component by almost all patients and GPs, which indicates that this NPI is well integrated into ambulatory CNCP therapy. To strengthen the care provider’s knowledge base regarding further guideline recommendations for the treatment of CNCP, the RELIEF intervention will provide respective e-learning modules to GPs and medical assistants. For GPs, this will include certified questions to acquire Continuing Medical Education points.

A key finding in this study is that most patients were willing to try out different options and engaged in NPIs such as physical activity, relaxation techniques, distraction, and stress reduction based on their own initiative, recommendations from their social environment, or from therapists, not their physicians. GPs perceived it was difficult for patients to integrate physical activity in daily routines and rather recommended physiotherapy and rehabilitation programs as offered by healthcare professionals. However, it remained unclear whether GPs actively sought a dialog with patients to arrive at this assessment. Some patients were eager to discuss a wider range of NPIs with their treating physicians, yet felt left alone with their pain. In contrast to findings from a study in 2010 where patients avoided discussing psycho-social aspects with their GPs for fear of stigmatization [38], patients in this study explicitly expressed a need for psychological coping interventions; one even concluded that the motivation to take actions against pain was strongly linked to psychological coping. This supports the recommendation that holistic and evidence-based pain management should not only aim to alleviate pain but also consider how patients cope with their pain situation [25]. Enduring pain, inadequate understanding of the disease concept, and avoidance of activities can lead to poor coping. Possible consequences are a strong protective behavior, excessive demands, or misuse of medication. Unsuitable coping strategies thus promote a negative development and further pain chronification [17]. This underlines the relevance of paying close attention to psycho-social treatment components and to a trusting physician–patient relationship as an important foundation of CNCP management [39]. A trusting relationship and educating patients are undisputedly central pillars of holistic pain management. As the first point of contact for all health issues, GP practices are an ideal place to provide basic information on the bio-psycho-social genesis of pain as well as pharmacological and non-pharmacological treatment options. This can be useful, both as initial information at the beginning of chronic pain symptoms and as a reinforcement, for instance, to promote the maintenance of learned coping strategies following rehabilitation or multimodal pain therapy [19]. Kaiser et al. [17] emphasize the need for an early start to therapy and a bio-psycho-social approach to pain treatment in primary care. The GP setting as chosen for the RELIEF intervention offers the potential to give more patients with CNCP access to sound pain management and improve their care. This poses major challenges beyond the gaps identified in this study and further research is needed to exploit this potential [19,25,40].

### Strengths and Limitations

Studies on the patient perspective regarding NPIs in CNCP treatment that included qualitative data are scarce. This study integrates both quantitative and qualitative data to inform intervention development in the project RELIEF. The strong interest of patients with CNCP to participate in this study illustrates the relevance of the topic. The purposive sampling ensured the inclusion of affected adult patients across all age groups. The reflexive analysis of the generated qualitative data facilitated the identification of relevant key and sub-themes across all interviews in a systematic yet flexible approach. Reporting of this study is based on scientifically valid criteria [41]. The individual telephone interviews facilitated comprehensive insights into personal experiences, perceptions, and attitudes, and accommodated for patient limitations due to pain or age, and the time resources of GPs, to minimize potential burden.

Some limitations must be mentioned. It is possible that individuals who were particularly interested in the topic and had a positive attitude towards the research project participated in the study. Selection bias is difficult to assess but cannot be excluded. No overall response rates were calculated since the percentage of affected individuals in the randomly contacted population sample is unknown, and the exact number of GP practices contacted via the DEGAM mail distribution list was not provided. Social desirability regarding statements made cannot be excluded. There is also a risk of interviewer bias and a risk of recall bias since the data were collected retrospectively. In order to counteract possible bias in the findings, the interprofessional project team discussed the individual work steps and data analysis. Patient participants had been dealing with CNCP for several years, which could introduce a recall bias, as they may have referred to past and current experiences with pain management, and not only to general practice care, but also to other disciplines and pain management facilities.

## 5. Conclusions

Patients with CNCP perceive bio-psycho-social influences on their pain and the need for holistic pain management. Integration of non-pharmacological interventions in the primary care setting and the physician–patient communication about them need strengthening. Structured pathways for outpatient CNCP care could facilitate improvements.

## Figures and Tables

**Figure 1 diseases-13-00034-f001:**
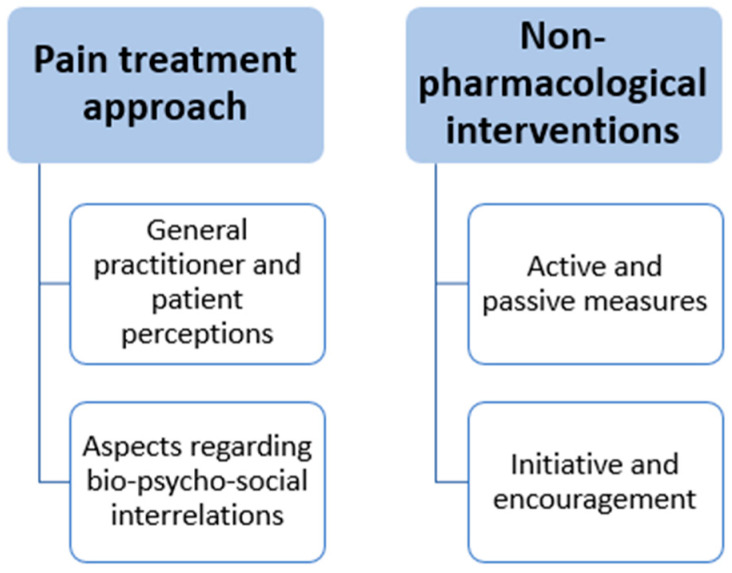
Identified key themes and sub-themes related to CNCP treatment and integration of non-pharmacological interventions.

**Figure 2 diseases-13-00034-f002:**
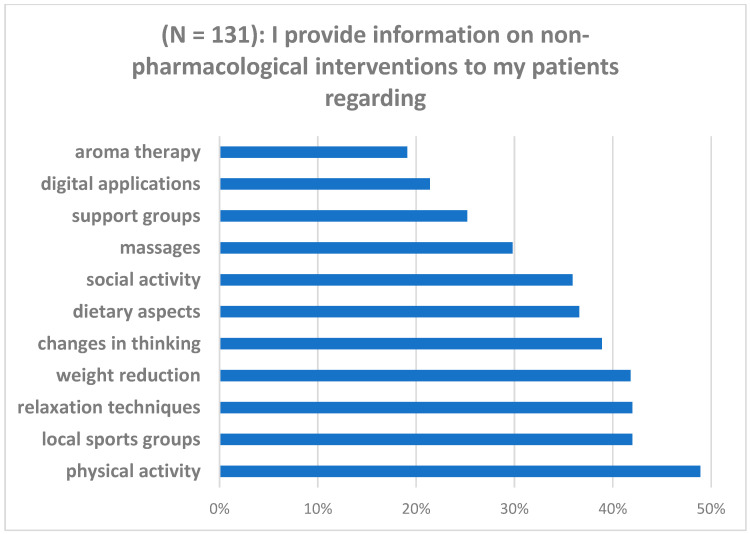
General practitioner perspective on verbally informing patients about non-pharmacological interventions for chronic pain.

**Figure 3 diseases-13-00034-f003:**
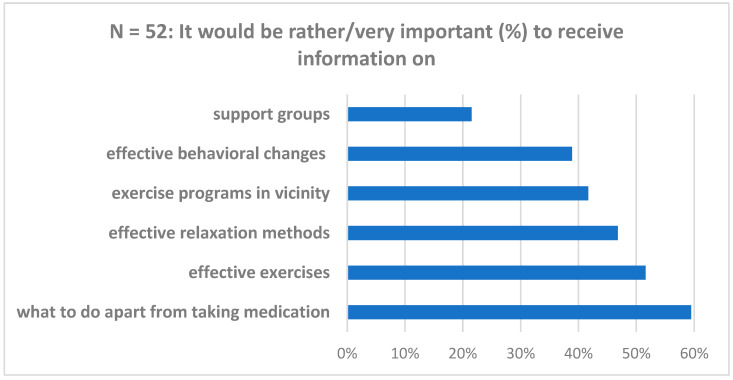
Patient perception regarding potential case management program components (n = 252).

**Table 1 diseases-13-00034-t001:** Socio-demographic participant characteristics.

	Survey	Interviews
**General practitioners** n	131	21
Gender f (%)	49 (49)	11 (52.4)
Age years mean (range)	48 (24–74)	50.6 (31–74)
Professional experience years mean (range)	20.5 (1–44)	21.3 (4–42)
Physician in specialization training in general practice * n (%)	8 (6.1)	2 (9.5)
Naturopathy specialist n (%)	33 (25.2)	4 (19.0)
Pain therapy specialists n (%)	26 (19.8)	2 (9.5)
**Patients** n	252	37
Gender f (%)	136 (54)	22 (58)
Age years mean (range)	62 (20–89)	69.5 (23–85)
Pain duration 0.5–20 years n (%)	230 (91)	37 (100)
Perceived pain intensity ** mean	5.5	6

* mandatory 5-year qualification program; ** scale 1–10 no pain–strongest pain.

**Table 2 diseases-13-00034-t002:** General practitioner perceptions regarding prevalence of chronic non-cancer pain and often-applied treatment approaches (n = 131).

Item	N *	“Often” %
In my practice, chronic non-cancer pain is a cause for consultation.	114	87.1
I ask patients what they do themselves to alleviate pain besides taking medication.	104	79.4
I make agreements with patients regarding integration of non-pharmacological interventions.	62	47.3
I provide information material regarding non-pharmacological interventions.	49	37.5
I refer patients to specialized pain therapy or outpatient clinics.	41	31.3
I use a structured approach with questionnaires or scales to taking individual medical pain history.	34	26.0

* excludes answers “seldom”, or not answered.

**Table 3 diseases-13-00034-t003:** Patient perceptions regarding disciplines involved in their chronic non-cancer pain treatment (n = 252).

Item	N *	%
I consulted a physician because of my pain during the last six months.	164	65.1
My General practitioner is involved in my pain treatment.	158	62.7
An orthopedist is involved in my pain treatment.	92	36.5
A physiotherapist is involved in my treatment.	67	26.6
A neurologist is involved in my pain treatment.	41	16.3
A pain therapy specialist is involved in my treatment.	18	7.1
A psychotherapist is involved in my treatment.	16	6.3
I go to an outpatient clinic.	3	1.2

* multiple answers were possible.

**Table 4 diseases-13-00034-t004:** Patient perceptions regarding discussed bio-psycho-social aspects of chronic pain (n = 252).

During the Course of My Pain Treatment, I Was Once Asked Whether	N	%
I have or had drug or alcohol related problems	139	55.2
I suffer from stress	137	54.4
I suffer from depression, anxiety and/or panic attacks.	136	54.0
there were traumatizing events in my past	136	54.0
I feel alone or lonely	134	53.2
I avoid certain activities for fear of pain	121	48.0
I am very afraid that my pain could get worse	117	46.4
I have difficulties with falling asleep or sleep through	108	42.9
I have support from friends and family	97	38.5

## Data Availability

All data generated and analyzed within this study are stored on a secure server at the Department of Primary Care and Health Services Research, University Hospital Heidelberg, Germany. De-identified sets of these data can be made available by the corresponding author on reasonable request.

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
