# Peer review of "Understanding General Practitioner and Patient Perceptions Regarding Integration of Non-Pharmacological Interventions in Chronic Non-Cancer Pain Management—A Cross-Sectional Mixed-Methods Study in the RELIEF Project"

_diseases, 2025, doi:10.3390/diseases13020034_

Round 1
Reviewer 1 Report
Comments and Suggestions for Authors
The manuscript reports a mixed-methods study gaining patient and general practitioner (GP) views around management of chronic non-cancer pain. Quantitative data were collected via a survey while qualitative data were collected through telephone interviews. The authors reported that GPs perceived they recommended non-pharmacological treatments in addition to pharmacological, but patients felt there was more emphasis on pharmacological. Patients were enthusiastic about non-pharmacological options, currently used them, and indicated a preference for more of these.
I like the way the program of research has been organized with the long-term view to developing and testing a new approach to managing chronic pain in the community. I think it is an important step to capture patient and GP views to inform development, so I think this is a worthwhile study.
The authors should be acknowledged for the extraordinary lengths they went to to recruit an appropriate sample of both patient and GP participants for the study. They made a particular effort to recruit a representative sample of patients based on age, gender, location, and citizenship. Often, those with more difficult access to technology and healthcare or those who are marginalized are left out of research, so I would like to commend the authors for their efforts here.
I think the authors could briefly describe and justify the methodological approach to the study, particularly the qualitative component. It would also be good to clarify if coding was done independently (i.e., a single interview was coded by multiple authors) or coding of the interviews was simply shared among the authors. I would also appreciate a little more explanation of the deductive component of the coding process. I am uncertain of what was involved in the sentence beginning “Based on the thematic focus in this study …” or what the “category system” refers to. I am also unsure on the key topics described as Pain treatment approach and Non-pharmacological interventions. Are these the themes developed? Or was this the deductive component with data coded within these pre-defined topics/themes?
I found that getting through the Results section was a little confusing jumping between the quantitative and qualitative findings and between patient and GP data. I think the Results section needs to be structured better and would suggest presenting all the quantitative findings together first, separated into the patient and GP results, then presenting the qualitative findings after that. As such, it was hard to see the themes in the qualitative component and how they sat together because it was interrupted by the quantitative findings.
The Discussion is fairly bland at present and covers the findings quite generally without delving into the related literature in much detail. I would like to see more discussion related to the purpose of the study around intervention development. For example, reconciling different patient and physician views, how these could be incorporated into the intervention, specific key learnings that are going to inform/have informed development of the intervention. This would keep the Discussion more focused and more specifically address the aim of the study.
The manuscript could do with a good English grammar check. It is not bad, and I understand that English may not be the authors’ first language, but there are a few instances where the wording could be improved for clarity and tidiness.
Specific comments
Page 2, lines 88-90. Does the current study reflect the “Prior to intervention development, previous patient experiences and GP perceptions of their own treatment approaches were explored” component mentioned in the preceding sentences? It would be good to clarify that and perhaps alter the wording a little to reflect this a little better. The topic is first raised in the past tense as though it has already been done, but from my understanding this manuscript reports on that process.
Page 4, line 168. I am not sure what the final part of this sentence refers to.
Page 4, lines 181-182. Were the quantitative data checked for normality prior to analysis?
Table 1. The pain experience variable is a little difficult to understand. Is this pain duration? Didn’t they all have to have pain for at least 3 months as part of the inclusion criteria?
Table 2. Were the GPs who did not see patients with chronic pain excluded from the remaining analyses?
Table 3. This looks the same as Table 2.
Author Response
Reviewer 1 comments:
The manuscript reports a mixed-methods study gaining patient and general practitioner (GP) views around management of chronic non-cancer pain. Quantitative data were collected via a survey while qualitative data were collected through telephone interviews. The authors reported that GPs perceived they recommended non-pharmacological treatments in addition to pharmacological, but patients felt there was more emphasis on pharmacological. Patients were enthusiastic about non-pharmacological options, currently used them, and indicated a preference for more of these.
I like the way the program of research has been organized with the long-term view to developing and testing a new approach to managing chronic pain in the community. I think it is an important step to capture patient and GP views to inform development, so I think this is a worthwhile study.
The authors should be acknowledged for the extraordinary lengths they went to to recruit an appropriate sample of both patient and GP participants for the study. They made a particular effort to recruit a representative sample of patients based on age, gender, location, and citizenship. Often, those with more difficult access to technology and healthcare or those who are marginalized are left out of research, so I would like to commend the authors for their efforts here.
Thank you very much for taking the time to assess our manuscript and provide recommendations to improve it. Your constructive feedback is highly appreciated. Please find our point-by-point answers below.
I think the authors could briefly describe and justify the methodological approach to the study, particularly the qualitative component. It would also be good to clarify if coding was done independently (i.e., a single interview was coded by multiple authors) or coding of the interviews was simply shared among the authors. I would also appreciate a little more explanation of the deductive component of the coding process. I am uncertain of what was involved in the sentence beginning “Based on the thematic focus in this study …” or what the “category system” refers to. I am also unsure on the key topics described as Pain treatment approach and Non-pharmacological interventions. Are these the themes developed? Or was this the deductive component with data coded within these pre-defined topics/themes?
Thank you for making us aware of these aspects. Yes, this refers to the deductive component with data coded within the pre-defined topics and identified themes. We added the missing information you referred to and re-phrased where applicable to increase transparency.
I found that getting through the Results section was a little confusing jumping between the quantitative and qualitative findings and between patient and GP data. I think the Results section needs to be structured better and would suggest presenting all the quantitative findings together first, separated into the patient and GP results, then presenting the qualitative findings after that. As such, it was hard to see the themes in the qualitative component and how they sat together because it was interrupted by the quantitative findings.
Thank you for pointing this out. To avoid confusing elements in the Results section, we followed your recommendation. Now, quantitative findings are presented first, followed by qualitative findings. However, we decided to keep the original structure of reporting about perceptions referring to treatment approaches first, followed by perceptions regarding non-pharmacological options.
The Discussion is fairly bland at present and covers the findings quite generally without delving into the related literature in much detail. I would like to see more discussion related to the purpose of the study around intervention development. For example, reconciling different patient and physician views, how these could be incorporated into the intervention, specific key learnings that are going to inform/have informed development of the intervention. This would keep the Discussion more focused and more specifically address the aim of the study.
Thank you also for this valuable recommendation. We agree that the discussion could be improved and amended it now. All changes are highlighted in the manuscript.
The manuscript could do with a good English grammar check. It is not bad, and I understand that English may not be the authors’ first language, but there are a few instances where the wording could be improved for clarity and tidiness.
Thank you for this recommendation. The manuscript was thoroughly checked for grammar mistakes again, both by the first author (bi-lingual) and a native English speaker. We are confident that we could improve wording to support clarity.
Specific comments
Page 2, lines 88-90. Does the current study reflect the “Prior to intervention development, previous patient experiences and GP perceptions of their own treatment approaches were explored” component mentioned in the preceding sentences? It would be good to clarify that and perhaps alter the wording a little to reflect this a little better. The topic is first raised in the past tense as though it has already been done, but from my understanding this manuscript reports on that process.
Thank you for pointing this out. To increase transparency, we changed the phrasing now.
Page 4, line 168. I am not sure what the final part of this sentence refers to.
Thank you again for pointing this out to us. We agree, the phrasing was not supporting clarity and we decided to delete the last part of the sentence.
Page 4, lines 181-182. Were the quantitative data checked for normality prior to analysis?
Thank you for this comment. Indeed, data were checked for plausibility, we added this information now.
Table 1. The pain experience variable is a little difficult to understand. Is this pain duration? Didn’t they all have to have pain for at least 3 months as part of the inclusion criteria?
Thank you for raising this question. You are absolutely right, this refers to pain duration and all participating patients had to have had pain for at least three months. The value provided in Table 1 indicates the experienced pain duration (6 months to 20 years) and the perceived pain intensity (scale from 1-10 no pain - strongest pain). We slightly changed the wording here to improve clarity.
Table 2. Were the GPs who did not see patients with chronic pain excluded from the remaining analyses?
Thank you for this comment which made us realize that we were not precise enough regarding the translation of this item and its’ intention, which was to ask about the frequency of CNCP as a cause for consulting with the GP. We changed the wording now to improve transparency.
All analyses of the survey data were performed with answered items only. Table 2 provides information on GP perceptions regarding prevalence of chronic non-cancer pain and often applied treatment approaches. Some participants answered ‘seldom’, or did not answer and are therefore not included in Table 2. We added a legend to the table to specify this.
Table 3. This looks the same as Table 2.
Thank you very much for pointing this out. We already had realized that Table 3 was not transferred properly onto the template, but could not amend this for reviewers. We apologize for the mix-up and have provided the correct table now.
Reviewer 2 Report
Comments and Suggestions for Authors
The theme is not novel however - important/
The study based on retrospective questinaries which have serious limitations
You must write that the study is retrospective
The problem if it is suitable in health service- the doctors are so busy and have time for not pharmacological treatment
It was written many times and well known
Author Response
Reviewer 2 comments and suggestions:
The theme is not novel however - important/
The study based on retrospective questinaries which have serious limitations
You must write that the study is retrospective
The problem if it is suitable in health service- the doctors are so busy and have time for not pharmacological treatment
It was written many times and well known
Thank you very much for taking the time to assess our manuscript. We also thank you for your recommendation regarding the limitations of the study and have added corresponding information now.
Also, we re-structured the Results section to provide better orientation regarding findings and data sources. In addition, some information was added to the Methods and Discussion sections. We trust that this eliminates your indicated concerns.
Reviewer 3 Report
Comments and Suggestions for Authors
The main conclusions seem to focus around differences between patient and physician views of how much discussion and understanding there is by GPs about the bio-psycho-social model of chronic pain. Further, the information about resulting recommendations or acceptability for non-medication interventions needs to be more clearly tied to what appears to be the overall theme and to whether the statement derive from the survey or the interview data. It suggests the title of the manuscript could be edited to focus on different understandings of the biopsychosocial model and how that is implemented in primary care.
Overall, the way the data were reported I found to be very confusing. Many statements are made throughout that don't seem to be directly linked to survey results or a summary of the interviews and at times the statements contradict each other. In several places survey data are described but not included in any table. See for example line 313 where "almost all patients were aware of the biopsychosocial model and line 450 where the same statement is attributed to 30% of patients. In addition, In several places survey data are described but not included in any table. Concepts are introduced without an accompanying data table and differentiation of survey versus interview data is unclear.
Here are a few additional specific points:
1) lines 142-143: what does "analysis of generated data" mean?
2) its not clear how practices were selected for interviews and it appears that perhaps more than one clinician was interviewed for some practices? Section 3.1 Sample Characteristics.
3) In Table 1 what is the meaning of "specialized training"? also there is a misaligned variable.
4) Table 3 proports to report patient data but it seems to be a repeat of GP data given in Table 2.
5) Section 3.2.3 reports various aspects of the biopsychosocial model but its not clear where these data come from since they are now shown in any table.
6) Line 294 the word "respectively" is used but only one finding reported.
7) Line 297 the term "already suffice" is unclear
8) Lines 323-325 I don't understand the meaning of the patient quote.
9) Section 3.3.2 top of p. 10 there is a discussion of the frequencies reported several sections earlier in Table 1 but there is no reference back to the table. Also it seems some items discussed in this paragraph don't appear int he GP survey chart.
10)Line 393 there is a reference to " all participating GPS" but it's not clear if these are survey or interview respondents.
11) Line 412, "most patients" -not clear if survey or interview data.
12)Discussion lines 457-473: It is unclear which statements refer to the references, which to survey data and which to GPS. the high rate of both patients and GPs endorsing physiotherapy is important but can't be traced back to presented data.
13) Line 508 says there was "strong interest" of patients to participate in the study which isn't supported by response rate etc.
Author Response
Reviewer 3 Comments and suggestions:
The main conclusions seem to focus around differences between patient and physician views of how much discussion and understanding there is by GPs about the bio-psycho-social model of chronic pain. Further, the information about resulting recommendations or acceptability for non-medication interventions needs to be more clearly tied to what appears to be the overall theme and to whether the statement derive from the survey or the interview data. It suggests the title of the manuscript could be edited to focus on different understandings of the biopsychosocial model and how that is implemented in primary care.
Overall, the way the data were reported I found to be very confusing. Many statements are made throughout that don't seem to be directly linked to survey results or a summary of the interviews and at times the statements contradict each other. In several places survey data are described but not included in any table. See for example line 313 where "almost all patients were aware of the biopsychosocial model and line 450 where the same statement is attributed to 30% of patients. In addition, In several places survey data are described but not included in any table. Concepts are introduced without an accompanying data table and differentiation of survey versus interview data is unclear.
Thank you very much for taking the time to assess our manuscript and for your recommendations. Thank you also for making us aware of these aspects, apologies for any confusion. To avoid confusing elements in the Results section, we now present quantitative findings first, followed by qualitative findings. We decided to keep the original structure of reporting about perceptions referring to treatment approaches first, followed by perceptions regarding non-pharmacological options. Not all reported quantitative data is presented in a table format, but under the respective header to ensure clarity about the source of the data. Regarding your comment referring to patient awareness of bio-psycho-social influences we would like to point out that the second value you mentioned (i.e. 30%) does not refer to our study, but is reported in a cited reference. We briefly added to the wording there to support transparency.
We thoroughly checked the manuscript for statements that potentially were not linked to presented data or were contradicting each other. We trust that the re-structuring of the Results section serves to eliminate your concern. Regarding the manuscript title, we took your remark into consideration. However, we decided to basically keep the original title with a slight adjustment since we feel that it reflects the aim of our study which was to explore the integration of non-pharmacological interventions into CNCP therapy from the perspective of patients and GPs (i.e. not the potentially different understandings of the bio-psycho-social model and its’ integration into primary care).
Here are a few additional specific points:
1) lines 142-143: what does "analysis of generated data" mean?
Thank you for drawing our attention to this aspect. The written consent encompassed consent to participate and consent for the scientific analysis and reporting of the collected data. To clarify and avoid confusion, we decided to change ‘generated’ to ‘collected’.
2) its not clear how practices were selected for interviews and it appears that perhaps more than one clinician was interviewed for some practices? Section 3.1 Sample Characteristics.
In section 3.1 we provide information regarding the sample characteristics. We did not select practices for an interview, we randomly selected publicly available addresses of GP practices in 71 communities in Baden-Württemberg in a web-based search and then contacted n=900 GP practices via e-mail. After expressing interest in participating in an interview, potential participants were sent an information leaflet and a declaration of consent. All GPs who returned the signed consent form, could subsequently be reached for appointment booking and could make time for the interview between June and November 2023 were interviewed. Only one GP per practice was interviewed. We added some information to sections 2.3 and 3.1. to increase transparency.
3) In Table 1 what is the meaning of "specialized training"? also there is a misaligned variable.
Thank you for drawing our attention to the misaligned variable in Table 1. We corrected the alignment now. Please accept our apologies, we cannot detect the phrase you referred to. However, the term “Physician in specialization training in general practice” used in in Table 1 refers to the mandatory 5-year training program physicians in Germany have to complete to fully qualify as General practitioner. We added some information to the table to increase transparency.
4) Table 3 proports to report patient data but it seems to be a repeat of GP data given in Table 2.
Thank you very much for pointing this out. We already had realized that Table 3 was not transferred properly onto the template, but could not amend this for reviewers. We apologize for the mix-up and have provided the correct table now.
5) Section 3.2.3 reports various aspects of the biopsychosocial model but its not clear where these data come from since they are now shown in any table.
We have changed the structure of the complete Results section now to provide a better overview of the findings and to minimize potential for confusion. All findings are now presented along the two key themes and under a header which indicates the data source (survey or qualitative data).
6) Line 294 the word "respectively" is used but only one finding reported.
We deleted the word to accommodate your concern.
7) Line 297 the term "already suffice" is unclear
We changed the wording here to increase clarity.
8) Lines 323-325 I don't understand the meaning of the patient quote.
Thank you for pointing this out. We also changed the wording to increase clarity now
9) Section 3.3.2 top of p. 10 there is a discussion of the frequencies reported several sections earlier in Table 1 but there is no reference back to the table. Also it seems some items discussed in this paragraph don't appear int he GP survey chart.
You are absolutely right, there was duplicate information without a reference to Table 1. We have corrected this now. You are also right, that the information in Figure 3 is not included in Table 1. Figure 3 provides details on non-pharmacological interventions GPs perceived to generally inform their patients with CNTP in verbal format, Table 1 refers to written material. We added information to clarify.
10)Line 393 there is a reference to " all participating GPS" but it's not clear if these are survey or interview respondents.
Apologies for any confusion. As we re-structured the entire Results section, this is corrected now.
11) Line 412, "most patients" -not clear if survey or interview data.
corrected now
12)Discussion lines 457-473: It is unclear which statements refer to the references, which to survey data and which to GPS. the high rate of both patients and GPs endorsing physiotherapy is important but can't be traced back to presented data.
We have added text here to increase clarity regarding your concerns. We also re-checked the manuscript to ensure that statements can be traced back to the presented data. The statements regarding physiotherapy you mentioned for instance trace back to 3.2.2 and 3.3.2 where findings of the qualitative data are presented. As mentioned before, we re-structured the Results section to increase transparency and reduce potential for confusion and trust this eliminates your concerns.
13) Line 508 says there was "strong interest" of patients to participate in the study which isn't supported by response rate etc.
Thank you also for this comment. We would like to point out that it was not known whether the randomly contacted individuals actually were suffering from CNTP. We received 252 completed patient survey questionnaires and 50 individuals indicated they wanted to participate in an interview. After screening for eligibility and scheduling efforts, we could interview 40 individuals. We do consider these numbers supportive for the statement.
Round 2
Reviewer 1 Report
Comments and Suggestions for Authors
Thank you for the opportunity to review the revised manuscript. The authors have done a great job responding to my previous comments and I think the manuscript is much improved.
I have only a couple of minor comments:
Lines 182-184. While the analysis of the qualitative data is described much better, I was still a little confused about this sentence. The previous sentence indicates that the data were coded and themes derived, but this sentence says the data were coded inductively again. I am not sure how the data can be coded again after themes were already developed.
Line 196, there is a word missing "along with themes"
Table 1, I suggest the * is used in the table before **
Line 368, I think this should be Table 2
Lines 505-508, perhaps indicate that how the findings helped to develop/shape the intervention are about to be discussed. When I first read this sentence I still wondered how they had specifically been used to guide intervention development, but this is explain more in the following paragraphs so I think it would be useful to indicate that overtly.
Author Response
Thank you for the opportunity to review the revised manuscript. The authors have done a great job responding to my previous comments and I think the manuscript is much improved.
Thank you very much for your assessment and your kind words.
I have only a couple of minor comments:
Lines 182-184. While the analysis of the qualitative data is described much better, I was still a little confused about this sentence. The previous sentence indicates that the data were coded and themes derived, but this sentence says the data were coded inductively again. I am not sure how the data can be coded again after themes were already developed.
Thank you Reviewer 1 for taking the time again to go through our manuscript and for your comment. You seem to refer to the following sentence: “Based on topics covered by the interview guides, a deductive step led to the final coding of the entire data.” We assume that you might have missed the word ‘deductive’ when reading this section. We slightly added to the text to clarify that the deductive step was conducted in addition to the inductive approach.
Line 196, there is a word missing "along with themes"
Thank you for pointing this out. We have added ‘with’ as suggested by you.
Table 1, I suggest the * is used in the table before **
Thank you for catching this. We changed that now and * is used before **.
Line 368, I think this should be Table 2
You are absolutely right, we correct this mistake.
Lines 505-508, perhaps indicate that how the findings helped to develop/shape the intervention are about to be discussed. When I first read this sentence I still wondered how they had specifically been used to guide intervention development, but this is explain more in the following paragraphs so I think it would be useful to indicate that overtly.
Thank you again for your recommendation. We agree with you and thus slightly added to the text.
Reviewer 3 Report
Comments and Suggestions for Authors
Authors have made a number of changes to the manuscript based on prior reviews. This has helped clarify which data came from surveys, which from interviews etc. Certain table errors have also been corrected although I should note that Table 1 under patient pain duration the first column states the range in years but the frequencies on the corresponding lines seem to be in months?
In the results, while some changes were made, it might be helpful to clearly identify which paragraphs relate to patient qualitative results and which to GP qualitative results, especially in sections 3.2.2 and 3.3.2.
Author Response
Authors have made a number of changes to the manuscript based on prior reviews. This has helped clarify which data came from surveys, which from interviews etc. Certain table errors have also been corrected although I should note that Table 1 under patient pain duration the first column states the range in years but the frequencies on the corresponding lines seem to be in months?
Thank you Reviewer 3 for taking the time again to go through our manuscript and for pointing us to this. We checked Table 1 and can confirm that it describes the number and percentage of participating patients who had a pain duration of 0.5 to 20 years (Pain duration 0.5 - 20 years n (%): Survey: 230 (91); Interviews: 37 (100)).
In the results, while some changes were made, it might be helpful to clearly identify which paragraphs relate to patient qualitative results and which to GP qualitative results, especially in sections 3.2.2 and 3.3.2.
Thank you very much for this suggestion which we carefully considered. We agree that respective sub-headings might be supportive for clarity in many cases. However, after inserting sub-headings into the text, we realized that this would be very repetitive. We thoroughly re-checked the text now and are confident that all paragraphs provide clear indication of the respective group the findings are related to and thus decided not to use extra subheadings.